# Novel Insights into the Differences in Nutrition Value, Gene Regulation and Network Organization between Muscles from Pasture-Fed and Barn-Fed Goats

**DOI:** 10.3390/foods11030381

**Published:** 2022-01-28

**Authors:** Yufeng Yang, Yan Wang, Huiquan Shan, Yalin Zheng, Zeyi Xuan, Jinling Hu, Mingsong Wei, Zhiqiang Wang, Qingyou Liu, Zhipeng Li

**Affiliations:** 1State Key Laboratory for Conservation and Utilization of Subtropical Agro-Bioresources, College of Animal Science and Technology, Guangxi University, Nanning 530004, China; yangyufeng@st.gxu.edu.cn (Y.Y.); wang9712yan@163.com (Y.W.); huiquan_shan@163.com (H.S.); yalin129@zyl.email.cn (Y.Z.); hujinling@st.gxu.edu.cn (J.H.); zhqwang@gxu.edu.cn (Z.W.); qyliu-gene@gxu.edu.cn (Q.L.); 2The Animal Husbandry Research Institute of Guangxi Zhuang Autonomous Region, Nanning 530010, China; xuanzeyi1@163.com (Z.X.); weims2022@163.com (M.W.)

**Keywords:** black goat, meat quality, pasture-fed and barn-fed, RNA-seq, lipogenesis

## Abstract

The physiological and biochemical characters of muscles derived from pasture-fed or barn-fed black goats were detected, and RNA-seq was performed to reveal the underlying molecular mechanisms to identify how the pasture feeding affected the nutrition and flavor of the meat. We found that the branched chain amino acids, unsaturated fatty acids, and zinc in the muscle of pasture-fed goats were significantly higher than those in the barn-fed group, while the heavy metal elements, cholesterol, and low-density lipoprotein cholesterol were significantly lower. RNA-seq results showed that 1761 genes and 147 lncRNA transcripts were significantly differentially expressed between the pasture-fed and barn-fed group. Further analysis found that the differentially expressed genes were mainly enriched in the myogenesis and Glycerophospholipid metabolism pathway. A functional analysis of the lncRNA transcripts further highlighted the difference in fatty acid metabolism between the two feeding models. Our study provides novel insights into the gene regulation and network organization of muscles and could be potentially used for improving the quality of mutton.

## 1. Introduction

Black goat (*Capra hircus*) has adapted to the heat and humidity of the sub-tropics, and Mashan black goat is one of the geographical indication varieties in Guangxi, China. The meat of black goat has become popular among consumers for its high-quality protein, low fat, and abundant trace elements. The quality of mutton from different geographical origins and feeding managements are varied, and the meat of the pasture-fed black goat is well-received for its unique nutritious quality and flavor in south China [1]. Consumers even tend to pay higher prices for the meat from pasture-fed black goats. Generally, meat from pasture-fed animals is regarded as more natural, and healthier in terms of rich omega-3 fatty acids, conjugated linoleic acid, and oleic acid, compared with that of barn-fed animals [2]. The exercise time and exercise intensity of pasture-fed goats are much higher than those of barn-fed goats, which may be one of the factors that leads to a better nutrition and flavor of pasture-fed goats [3]. Moreover, the feeding cycle, feed composition and total energy intake in pasture-fed goats are also widely different with barn-fed goats, which results in a different of growth rate, fat deposition, and fatty acids constituent in the meat [4]. Although the meat of pasture-fed goats is widely recognized as highly nutritious and flavorful, the underlying molecular mechanism identifying how the pasture feeding affects the nutrition and flavor of the meat is still not fully revealed. 

Myogenesis is an extremely complex signal regulatory network, mediated by diverse molecules [5]. The nutrition or feeding regimen can regulate the expression of skeletal muscle growth-related genes, such as the myogenic regulatory factors (MRFs) [6] and myocyte enhancer factor-2 (MEF2) [7]. Finally, it can affect the skeletal muscle fiber type and composition during skeletal muscle growth. Researchers have found that pasture-feeding has positive effects on meat color [8] and nutritional attributes, through its effect on the fatty acid profile of meat lipids [9]. Intramuscular fat deposition, fatty acid content and type make up the main factors that affect the quality of goat meat; intramuscular fat (IMF) has a positive effect on the sensory quality traits of meat, including the flavor, juiciness, and tenderness [10]. The regulation of intramuscular fat deposition and metabolism needs to go through complex biological processes, including the transport of fat to muscle cells, the binding and transport of fat in cytoplasm and organelles (mitochondria and endoplasmic reticulum), and the regulation of triacylglycerol synthesis and decomposition in muscle [11,12]. Numerous genes, including fatty acid synthase (FASN) [13], acetyl-CoA carboxylase alpha (ACACA) [14], glycerol-3-phosphate acyltransferase (GPAT) [15], stearoyl-CoA desaturase (SCD) [16] and peroxisome proliferator-activated receptor gamma (PPARg) [17], have been found to playing crucial roles in the adipogenesis. In addition, lncRNAs are also considered to play key roles during various biological processes by acting on the target genes [18,19]. The genome-wide identification of LncRNAs in the skeletal muscle of fetal goat found that numerous LncRNAs were functionally enriched in transcriptional regulation and development-related processes [20]. LncR-125b can promote the differentiation of skeletal muscle satellite cells by functioning as a competing endogenous RNA (ceRNA) for miR-125b to control insulin-like growth factor 2 (IGF2) expression [21]. The knockdown of lncIMF4 in pigs promotes adipogenesis by attenuating autophagy to repress the lipolysis in intramuscular adipocytes [22]. LncSAMM50 is highly expressed in adipose tissue, and can inhibit the adipogenic differentiation in 3T3-L1 cells [23]. Therefore, study on the gene expression patterns in the meat from pasture-fed and barn-fed goats should provide important information about the underlying molecular mechanism of development and fat deposition in the muscle. 

This study aimed to systemically reveal the differences in the physiological and biochemical characters of longissimus lumborum derived from pasture-fed or barn-fed black goats, and explore the genetic basis of the high-quality meat from pasture-fed black goats using RNA-sequence. The current study can provide novel understanding about ways to improve feeding management in black goats, to the benefit of improving the product efficiency and meat quality in mutton production. 

## 2. Materials and Methods

### 2.1. Sample Preparation

All animals received humane care as outlined in the Guide for the Care and Use of Experimental Animals of the National Institutes of Health. The animal experiments were approved by the Animal Experiments Ethical Review Committee of the Guangxi University, Nanning, China (Grant NO.: Gxu-2021-158). The longissimus lumborum samples (between the 12th and 13th thoracic vertebrae) were obtained from 6 black goats (12-month-old castrated rams) in a local commercial abattoir in Mashan, Guangxi province. The barn-fed goats were fed in attic houses with a double-pitched roof. The leaky floor was more than 1.8 m above the ground. There was a rear window in every 100 m^2^ of the house, and the total daylighting area of the window was more than 1/20 of the total area. Each goat occupies an average space of 0.8–1.2 m^2^. As this research was performed in winter, the temperature in the goat house was about 10–25 °C, the humidity was about 60~70%, and the illumination time was about 10 h. For the pasture-fed goats, they were fed in a natural environment where they could move and seek food and water freely. We randomly selected 3 goats from the barn-fed and pasture-fed group, respectively, for the following study. 

### 2.2. Meat Quality Evaluation 

To analyze the characteristics of muscle fibers, the fresh longissimus lumborum tissues were fixed with paraformaldehyde and gradient-dehydrated with sucrose solution. Frozen sections were made and stained by eosin methylene blue. The muscle samples were obtained immediately after the slaughtering and the meat color data were read with the meat color meter (MATTHAUS, Neu-Isenburg, Germany). The muscle conductivity was detected using the carcass meat conductivity tester LF-STAR (MATTHAUS, Neu-Isenburg, Germany). Water content was determined on drying at 100 °C for 24 h. The cooking loss was measured by weighing the steaks before and after cooking. The shear force was measured with C-LM3 digital display muscle tenderness instrument (MATTHAUS, Neu-Isenburg, Germany) according to the instrument’s instructions. In brief, the muscle samples with length × width × height more than 6 cm × 3 cm × 3 cm were prepared, and the fascia and fat were removed. The samples were then put into 80 °C water bath until the center temperature reached 70 °C. The shear force was measured after cooling with cold water. The fresh meat was placed at 4 °C for 24 h, and the pH 24 was measured using a pH meter (Thermo Orion, Hudson, NH, USA) with potassium hydrogen phthalate solution (pH = 4.0) and a mixed phosphate buffer (pH = 6.8) as the calibration buffers. The crude protein was determined by Kjeldahl nitrogen analyzer (Flowserve Sweden, Kjeltec 8400, Hillerod, Denmark), the content of crude ash was evaluated by ashing at 600 °C for 10 h, and the crude fat was obtained by petroleum ether extraction. Amino acid composition and content were determined using the amino acid analyzer (Hitachi High-Tech Corporation, LA8080, Tokyo, Japan), and the mineral content was determined by inductively coupled plasma–mass spectrometry (ICP-MS) (Thermo Fisher Scientific, iCAP Q, Waltham, MA, USA). The determination of fatty acids was carried out in accordance with the national food safety standard determination of fatty acids in foods (GB 5009.168-2016) issued by the State Food and Drug Administration (http://down.foodmate.net/wap/index.php?moduleid=23&itemid=50488 accessed on 17 December 2021). In brief, the muscle sample was firstly hydrolyzed by hydrochloric acid, and the fat extract was collected. The fat was then saponified and the fatty acid was methylated; the fatty acid was finally determined by gas chromatography (Shimadzu, GC-2010, Kyoto, Japan). The calculation of Health Lipid Indices refers to the previous studies [24]: NVI (Nutritive Value Index); AI (Atherogenic Index); TI (Thrombogenic Index); OFA (undesirable fatty acids); DFA (dietary fatty acids). Serum biochemical indexes were measured by the automatic animal biochemical analyzer (URIT, U-8020, Guilin, China). At least 3 samples were used in each test and each sample was detected more than 3 times.

### 2.3. Library Preparation and RNA Sequencing 

The total RNA was extracted from the longissimus lumborum samples using TRIzol reagent (Invitrogen Life Technologies Inc., Carlsbad, CA, USA) according to the manufacturer’s instructions. The genomic DNA eliminated by digestion with DNase I (Thermo Scientific, Waltham, MA, USA), the RNA quality and quantity were determined using NanoDrop 2000 (Thermo Scientific). The rRNA was removed with the probe, and then the remaining RNA was used for library construction and sequencing (ribo-zero RNA SEQ). The preparation of the cDNA library and Illumina sequencing analysis were performed referring to a previous study [25]. 

### 2.4. Lncrnas Identification

LncRNA was screened by the following steps: (1) we selected the transcripts with the following characteristics indicating possible pre-mRNA fragments: predicted transcript fell entirely within a reference intron; exon of predicted transcript overlapped with the reference but lay on the opposite strand; predicted transcript was intergenic in comparison with known reference transcripts; exon of predicted transcript overlapped with a reference transcript; predicted single-exon transcript overlapped with a reference exon plus at least 10 bp of a reference intron. (2) We selected transcripts with a length ≥200 bp and number of exons ≥2; (3) we selected transcripts with fragments per kilobase of transcript per million fragments mapped (FPKM) ≥0.1; the above candidate lncRNA were further screened, mainly including Coding Potential Calculator (CPC) analysis, Coding-Non-Coding Index (CNCI) analysis, Coding Potential Assessment Tool (CPAT) analysis, and Pfam protein domain analysis. 

### 2.5. Gene Ontology and Pathway Analyses

The FDR < 0.05 and |log2 (fold change)| > 1.5 values were used as a criterion to find the differentially expressed genes (DEGs). The DEGs were enriched and analyzed by ClusterProfiler (R version 3.5.1, University of Auckland, Auckland, New Zealand), Gene Ontology (http://www.geneontology.org; accessed on 5 April 2021). GO cluster analysis of differentially expressed lncRNA CIS target genes pooled the GO terms corresponding to all differential genes at the third classification level, calculated the overall expression level of each term in each sample, and conducted go functional cluster analysis. The pathway annotation of differentially expressed genes was analyzed using the KEGG database (Kyoto Encyclopedia of Genes and Genomes), KEGG pathway (http://www.kegg.jp; accessed on 15 April 2021).

### 2.6. Quantitative Reverse Transcription-Polymerase Chain Reaction (qRT-PCR) Analysis

The total RNA of longissimus lumborum of black goats in the pasture-fed group and barn-fed group was reverse transcribed with Hiscript RⅡ of RT-PCR kit (Vazyme, Nanjing, China). The primers were designed using Oligo software (version 7.56, DBA Ol-igo, Inc., Vondelpark Colorado Springs, CO, USA) (Appendix A) and synthesized by GenSys Biotech (Nanning, China). qRT-PCR was performed using the SYBR qPCR Master Mix (Vazyme, Shanghai, China) following the manufacturer’s instructions. Fluorescence data were acquired using the fluorescence ratio PCR instrument (Roche, Shanghai, China). More than 3 biological replicates and technical replicates were performed. The relative gene expression was calculated using the 2^−^^△△CT^ method [26], with β-actin as the reference gene. 

### 2.7. Statistical Analysis

Statistical analysis was conducted by using Student’s *t*-test and analysis of variance (ANOVA) with DUNCAN’s Multiple Range Test (DMR) in SPSS software (version 17.0, IBM, Armonk, New York, USA). Data were expressed as mean ± SEM, and *p* < 0.05 was considered as statistically significant.

The study was conducted in accordance with the Declaration of Helsinki, and approved by the Animal Experiments Ethical Review Committee of the Guangxi University, Nanning, China (protocol code Gxu-2021-158 and 12 November 2020).

## 3. Results

### 3.1. Differences of Physiological and Biochemical Characters between Pasture-Fed and Barn-Fed Goat Meat

The physiological and biochemical characters of longissimus lumborum derived from pasture-fed or barn-fed black goats were analyzed and compared. More than 100 muscle fibers in each muscle sample were measured and analyzed in the frozen section, and the results showed that the area of the longissimus lumborum of pasture-fed black goats was significantly smaller than that of barn-fed black goats (Figure 1A; Appendix A). The pH, meat color, and the cooking loss of black goats in the muscle of pasture-fed group were found to be significantly higher than those in the barn-fed group, while the electrical conductivity and crude fat in the muscle was significantly higher in the barn-fed goats (Figure 1B; Appendix A). In addition, we also systematically detected and analyzed the amino acid composition, fatty acid composition and the mineral content in the muscle. Results showed that the essential amino acids (such as isoleucine and leucine) in the muscle of the pasture-fed group were significantly higher than those in the barn-fed group (Figure 1C; Appendix A). The total fatty acid content and saturated fatty acid ratio in pasture-fed meat were significantly lower than those in the barn-fed meat (Table 1), indicating that the pasture-fed meat is more beneficial to human health. However, the higher crude fat content and fibrous area suggest a better taste in the barn-fed meat. Based on the above data, we found that the nutritional value index (NVI) of pasture-fed goat muscle was significantly higher than that of barn-fed. In addition, the OFA index, which describes dietary fatty acids that have adverse hypercholesterolemic effects on humans, was significantly lower in the pasture-fed goat muscle (Table 1), suggesting that the muscle of the grazing group is more beneficial to human cardiovascular health. We also found that there were more wholesome mineral elements, such as zinc and magnesium, and fewer heavy metal elements, including manganese and arsenic, in the meat of pasture-fed meats. Plumbum was not detected in either of the muscles (Figure 1D; Appendix A). We further analyzed the serum biochemical indexes of the goats and the results showed that the total cholesterol and low-density lipoprotein cholesterol (LDL-C) in the serum of the pasture-fed group were significantly lower than those in the barn-fed group (Figure 1E; Appendix A), which further indicates that eating muscle from a pasture-fed goat is more beneficial to human cardiovascular health.

### 3.2. Transcriptome Differences between Muscles of Pasture-Fed and Barn-Fed Black Goats

The total RNA was extracted and sequenced to reveal the different gene expression pattern between the meat from pasture-fed or barn-fed black goats. Hierarchical cluster analysis was performed to show the expression variation and cluster character of the genes (Figure 2A). A total of 22,775 genes were detected, of which 12,760 genes were detected in both groups, 349 genes were specifically detected in the barn-fed, and only 5 genes were specifically detected in the pasture-fed group (Figure 2B). Analyzing the gene expression data, 1761 genes were found to be significantly differentially expressed (Appendix A), including 878 up-regulated and 883 down-regulated genes (Figure 2C,D). We paid particular attention to the genes related to muscle development and lipogenesis, and found that the expression of myocyte enhancer factor 2C (MEF2C), myostatin (MSTN), FASN, 3-hydroxy-3-methylglutaryl-CoA synthase 1 (HMGCS1), low-density lipoprotein receptor (LDLR), farnesyl-diphosphate farnesyltransferase 1 (FDFT1), 7-dehydrocholesterol reductase (DHCR7), acyl-CoA synthetase short chain family member 2 (ACSS2), phosphate cytidylyltransferase 1A (PCYT1A), SCD, and peroxisome proliferator activated receptor alpha (PPARα) were significantly up-regulated in the barn-fed group. While the expression of acyl-CoA synthetase long chain family member 5 (ACSL5), lipin 2 (Lpin2) and perilipin 2 (Plin2) were significantly down-regulated in the barn-fed group. The top 30 of the differentially expressed genes (DEGs) were further analyzed by Gene ontology (GO) clustering and the results showed that 8 or 2 of them were enriched in locomotory behavior and feeding behavior-related terms, respectively (Figure 2E). Kyoto Encyclopedia of Genes and Genomes (KEGG) analysis was also performed, and it was found that the DEGs were mainly enriched in Glycerophospholipid metabolism, AMPK signaling pathway, and insulin signaling pathway (Figure 2F). 

### 3.3. Identification of lncRNAs in Goat Muscles

The LncRNAs were extracted from the transcriptome profiles using a highly stringent filtering pipeline, as shown in the method. The intersection of the Coding Potential Calculator (CPC) analysis, Coding-Non-Coding Index (CNCI) analysis, Coding Potential Assessment Tool (CPAT) analysis, and Pfam protein domain analysis results finally yielded 5122 lncRNA transcripts (Figure 3A, Appendix A), including 2895 lincRNAs, 1495 antisense-lncRNA, 670 intronic-lncRNA and 65 sense-lncRNA (Figure 3B). The ORF of the mRNA and screened lncRNA were predicted, and the results showed that the ORF length of lncRNA was significantly shorter than that of mRNA (Figure 3C). Comparison of the exon features of the lncRNAs and mRNAs showed that most lncRNAs contained 2–5 exons per transcript (about 3.3, on average), which was fewer than the mRNAs (Figure 3D). Similar to mRNAs, lncRNA transcripts were distributed widely in the chromosomes (Figure 3E). The potential target genes of the lncRNAs in cis-regulatory relationships were predicted to investigate the possible functions, and 4697 lncRNAs were found transcribed close to (<10 kb) protein-coding neighbors.

### 3.4. Differentially Expression and Enrichment Analysis of lncRNAs

The expression levels of the lncRNA transcripts from pasture-fed and barn-fed goat muscle were also analyzed, and 147 differentially expressed lncRNA transcripts were identified (Appendix A), including 111 up-regulated and 36 down-regulated lncRNAs (Figure 4A). To explore the similarities and relationships between the different libraries, the expression of the differentially expressed lncRNAs were measured using systematic cluster analysis (Figure 4B). Moreover, diverse lncRNAs were found to target genes playing crucial roles in the synthesis of fatty acids and unsaturated fatty acids, including FASN, LDLR, FDFT1, HMGCS1, ACSS2, scavenger receptor class B member 1 (SCARB1) and SCD (Figure 4C). GO analysis showed that muscle myosin complex, muscle contraction, and muscle cell differentiation terms were significantly enriched (*p* < 0.05) (Figure 4D), indicating that different locomotory behavior between the two groups influenced the transcript of the lncRNAs. Interestingly, KEGG analysis showed that lncRNAs, including MSTRG.17648.1, MSTRG.41627.1, MSTRG.57988.1, MSTRG.57988.4, MSTRG.19078.1, that were annotated with non-alcoholic fatty liver disease (NAFLD) (Figure 4E), suggesting that the transcriptional regulation of target genes may be one of the main effects of pasture-fed feeding, which finally affects the nutrient content during muscle development.

### 3.5. Analysis of the Underlying Molecular Mechanism and Validation of Differentially Expressed Genes and lncRNAs

To further reveal the underlying molecular mechanism of different crude fat and UFA content caused by different feeding patterns, 15 of the differentially expressed genes and 8 lncRNAs related to the fat biological process or muscle development were drawn out. The genes and lncRNAs were mainly found to be enriched in the myogenic regulatory factors, the cholesterol synthesis pathway, fatty acid synthesis, the degradation pathway, and the choline metabolism pathway (Figure 5A). Among them, the expression of MSTN, myosin heavy chain 7B (MYH7B), and MEF2C were found to be significantly up-regulated in the pasture-fed group. The up-regulated expression of the MSTN gene, which encodes a protein that negatively regulates skeletal muscle cell proliferation and differentiation, may lead to the lower carcass weight in the pasture-fed goats. MYH7B and MEF2C are genes related to the formation of slow-twitch myosin. Therefore, the up-regulated expression of MYH7B and MEF2C suggested that the development of slow-twitch fibers in the muscle of pasture-fed group was better. The expression of PLIN2, which is an important gene in the synthesis of triglycerides, was significantly down-regulated in the barn-fed group. However, the expression of SCD and PPARα, which can promote the synthesis of UFA, was significantly up-regulated in the barn-fed group. The expression of genes that beneficial to the reduce of serum cholesterol, such as LDLR, HMGCS1, DHCR7 and FDFT1, are also significantly up-regulated in the barn-fed group, which may be the factor that led to the lower cholesterol in the serum of barn-fed goat. qRT-PCR was further performed to validate the expression of the selected genes in the muscles from the two groups. Results showed that the expression patterns of 13 genes were consistent with those from the RNA-seq data. However, the expression pattern of FDFT1 was contrary to the RNA-seq data, and no significant difference was found in the expression of LDLR (Figure 5B). While for lncRNAs, the expression pattern of the tested lncRNAs was found to be consistent with the RNA-seq data (Figure 5C). The results showed that the different feeding model may have caused the different expression of the genes and lncRNAs related to muscle development and lipid metabolism. In addition, the combined action of the above molecules may have led to the increase in cholesterol and crude fat synthesis in barn-fed goat meat, which is consistent with the physiological and biochemical results.

## 4. Discussion

In this study, we systemically analyzed the physiological and biochemical characters of longissimus lumborum derived from pasture-fed or barn-fed black goats and found that the pasture-fed mutton is more beneficial to human health, containing more UFA and zinc, and fewer heavy metal elements and selenium. The content of branched chain amino acids, which are main factors that affect the flavor of meat quality, such as Leu and Ile, were found to be significantly higher in the pasture-fed group than those in the barn-fed group, revealing the basis of the flavor of pasture-fed goat meat. RNA-seq found that 1761 genes were significantly differentially expressed, including 878 up-regulated and 883 down-regulated genes. GO and KEGG analysis showed that the differentially expressed genes were mainly enriched in myogenesis and Glycerophospholipid metabolism pathway, presenting the underlying gene regulation and network that affects the fatty acid metabolism in the mutton. We also identified 5122 lncRNA transcripts from the RNA-seq data, and found 147 lncRNA transcripts were significantly differentially expressed in the two feeding models. Diverse target genes that played crucial roles in the synthesis of fatty acids and unsaturated fatty acids were annotated in the lncRNA transcripts, including FASN, LDLR, FDFT1, HMGCS1, ACSS2, SCARB1, SCD and 7 NAFLD-related genes, which further highlights the difference in fatty acid metabolism between the meat from pasture-fed or barn-fed black goats. The potential function of the expression pattern of the genes and lncRNAs was consistent with the data of physiological and biochemical characters, which revealed the underlying molecular mechanism that led to the differences between the pasture-fed and barn-fed goat muscle.

The feeding model is one of the main factors affecting the quality of mutton. The edible quality of mutton mainly depends on taste, followed by tenderness and juiciness [27], though great differences in preferences exist according to different cultural and cooking methods [28]. Research has found that, in terms of taste, smell and tenderness, forage-fed lambs have greater variability than those fed in fences [29]. The flavor of the meat is mainly affected by the metabolite of amino acids in the muscle. For example, branched chain amino acids (Leu, Ile, Val) produce malt, fruit and sweet flavor, aromatic amino acids (Phe, Tyr, Trp), flower and chemical fragrances, Aspartic acid (ASP), and a butter flavor [30]. Therefore, the richer Leu and Ile in the meat from pasture-fed goats may be the key reason for the better flavor. In addition, pasture-fed has been proved to have a positive impact on the image and nutrition of goat meat [31], and flesh color is closely related to pH [32]. The color of pasture-fed meat is commonly deeper for a high content of myoglobin and the content of intramuscular fat is lower, which were also proven in our study. In this study, although the higher crude fat content and fibrous area suggested a better taste in the barn-fed meat, the higher UFA in the pasture-fed meat seemed to be better for human health. Moreover, the richer zinc and magnesium content, the significantly higher nutritional value index, and lower OFA index in the pasture-fed meat, further suggest the benefit to human cardiovascular health. 

The development of skeletal muscle is a highly coordinated process that involves an extremely complex signal regulatory network. Diverse molecules, including genes, miRNAs and lncRNAs, have been found to participate in skeletal muscle growth via multiple signal pathways and in various manners. For example, Myogenic regulatory factor (MRFs) [6], paired box protein 3/7 (Pax3/7) [33], Myostatin (MSTN) and myocyte enhancer factor 2 (MEF2) [7] family are key factors that regulate muscle development. Meanwhile, these factors can also be regulated by lncRNAs and proteins. Myosins are hexamers composed of two myosin heavy chains (MHCs), and four myosin light chains (MLCs) [34]. MHCs, which are encoded by the MYHs gene, are considered as the molecular markers of muscle fiber type [34]. MRFs control the differentiation of skeletal muscle cells during myogenesis, and the expression of Myf5 and Myf6 were remarkably affected by feeding system [35]. A study has found that the expression of Myf5 and Myf6 in a barn-fed group was greater than in grazing goats, indicating that their mRNA expression is correlated with nutrition and activity levels [35]. However, no significant difference of the expression of Myf5, Myf6, MyoG and MyoD were found between the pasture-fed and barn-fed groups in our study. Instead, the expression of MYH7B, which is a slow-twitch myosin [36], was found to be up-regulated in the pasture-fed group. We speculate that this inconsistency may be due to the differences in nutrition and exercise levels between the previous study and our study. For example, the different terrain means different exercise levels in the pasture-fed group, and different feed formulations led to the different nutrition in the barn-fed group. The expression of MEF2s is initiated by MRFs, and is essential for the specific gene expression of slow-twitch fibers in muscles [37]. In the current study, the expression of MYH7B and MEF2C was significantly up-regulated in the pasture-fed group, suggesting the development of slow-twitch fibers in the muscle of the pasture-fed group was better. This is consistent with the fact that the pasture-fed group requires more constant movement. The expression of MSTN in the pasture-fed group was also up-regulated in our study, indicating the development of muscle undergoing more inhibiting effects. This result also revealed the reason for the lower carcass rate of pasture-fed goats.

The feeding cycle, total energy intake, growth rate, fat deposition and composition of fatty acid ratio are the main reasons why different feeding methods affect different flavors and nutritional levels [31]. The number of muscle fibers remains unchanged after birth, while adipose deposited in the muscle increase with age [35]. The content of intramuscular fat deposition is an important indicator of muscle quality, while the composition of the fatty acids in the meat seems to be more important for human health. Fatty acid synthesis and metabolism can be affected by feeding models [4], as pasture-fed means more physical activity and barn-fed means more grain feed intake. The level of physical activity is the main determinant of regulating fatty acid oxidation and adapting to lipid and glucose supply [38]. During exercise, energy homeostasis is mediated by signaling molecules between the liver, adipose tissue and skeletal muscle, and moderate-intensity exercise causes changes in energy metabolism, leading to the production of fatty acids [39]. Using RNA-seq, we found that the differentially expressed genes and lncRNAs between muscles from pasture-fed and barn-fed goats were mainly enriched in the cholesterol and triglyceride synthesis pathway. 3-hydroxy-3-methylglutaryl-CoA synthase 1 (HMGCS1) [40] and 7-dehydrocholesterol reductase (DHCR7) are key steps in cholesterol synthesis [41], and a low-density lipoprotein (LDL) receptor is the main media mediation for the input of cholesterol [42]. Squalene synthase (SQS), encoded by FDFT1, is also a key regulator of cholesterol synthesis [43]. In this study, the expression of these genes was up-regulated in the meat from barn-fed goats, which revealed the molecular mechanism of the richer cholesterol in the meat and serum of the barn-fed goats. In addition, the genes associated with fatty acid syntheses, such as FASN and ACSS2, were also up-regulated in the meat from barn-fed goats. Genes that can promote the synthesis of UFA, such as SCD and PPARα, were significantly up-regulated in the meat from pasture-fed goats. These genes, together, mean that barn-fed muscle contains more fatty acids, while the pasture-fed muscle contains more UFA, which is consistent with the results of the biochemical test. 

In addition, though the mechanism was not fully revealed, lncRNAs have been considered to play key roles in muscle development [20] and fat deposition [44] by regulating the expression of target genes. Studies have shown that lncRNAs acted as essential regulators in myogenesis and adult skeletal muscle regeneration. For example, noncoding RNA steroid receptor RNA activators (SRA), MUNC and LncMyoD, have been reported to promote myogenic differentiation by regulating the transcriptional activity of MyoD [45]. lnc-MD1, Glt2/Meg3, lnc-YY1 and lncRNA-Dum21 are also believed to be important positive regulators of myogenesis [46]. In contrast, m½-sbsRNA, Yam (YY1-associated muscle lncRNAs) and lnc-31 were reported to inhibit the myogenic differentiation [46]. However, in this study, no significant different expression of lncRNAs that target MRFs, MEF2s or MSTN genes was found between the muscle from the pasture-fed and barn-fed goats. Interestingly, diverse lncRNAs that target genes related to NAFLD were identified in our study. NAFLD is caused by a build-up of fat in the liver and mainly occurs in people who are overweight or obese [47]. The target genes of the lncRNAs are mainly enriched in the Oxidative phosphorylation, EGFR tyrosine kinase inhibitor resistance and MAPK signaling pathway, which is related to diverse functions such as energy metabolism and adipogenesis. Scavenger receptor Class B Type 1 (Scarb1) is a major receptor-mediating cholesterol transfer to and from high density lipoprotein cholesterol (HDL). We found that the expression of MSTRG.39270.1 which targeted Scarb1 was significantly down-regulated in the pasture-fed group, which is consistent with the result that less cholesterol was detected in the serum of pasture-fed group.

## 5. Conclusions

Our study showed that the minerals, amino acids, fatty acids and serum biochemical indexes of muscles derived from pasture-fed goats were significantly different from barn-fed black goats. By RNA-seq analysis, we further identified diverse candidate genes and lncRNAs that may be affected by the different feeding model, and provided valuable insight into the differences in gene regulation and network organization in the two feeding models. The genes and pathways associated with the two feeding models revealed here are candidates for follow-up validation studies, and could potentially be used for designing a breeding program to genetically improve the quality of mutton in the future.

## Figures and Tables

**Figure 1 foods-11-00381-f001:**
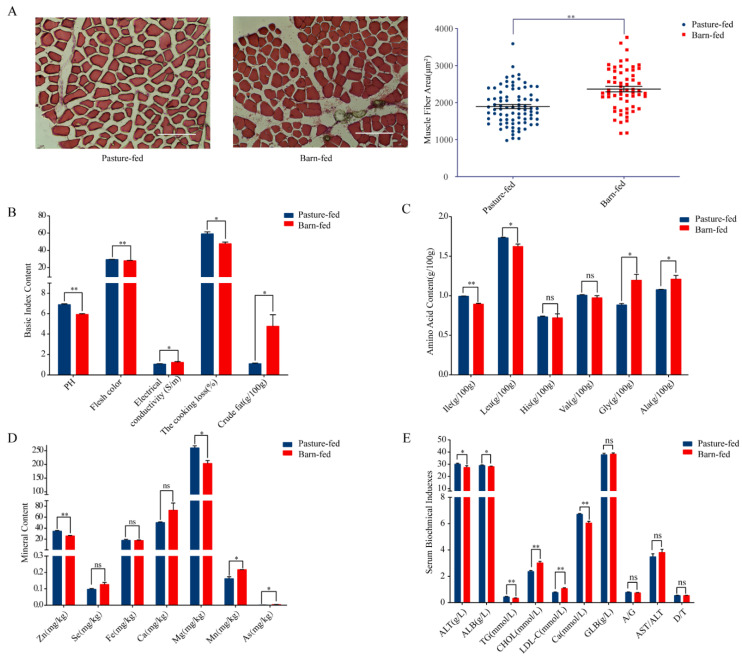
Physiological and biochemical characters in longissimus lumborum from pasture-fed and barn-fed goats. (**A**) Histological frozen section to analysis cross-sectional area of longissimus lumborum. (**B**) Basic nutrition index content of muscle from pasture-fed and barn-fed goat meat. (**C**) The amino acids content with significant differences between pasture-fed and barn-fed goat meat. (**D**) Mineral content in the pasture-fed and barn-fed goat meat; (**E**) Serum biochemical indexes of the pasture-fed and barn-fed goat meat. ALT: glutamic-pyruvic transaminase; ALB: serum albumin; GLB: seroglobulin; D/T: direct bilirubin/Total bilirubin. * and ** indicate significant different (*p* < 0.05 or *p* < 0.01).

**Figure 2 foods-11-00381-f002:**
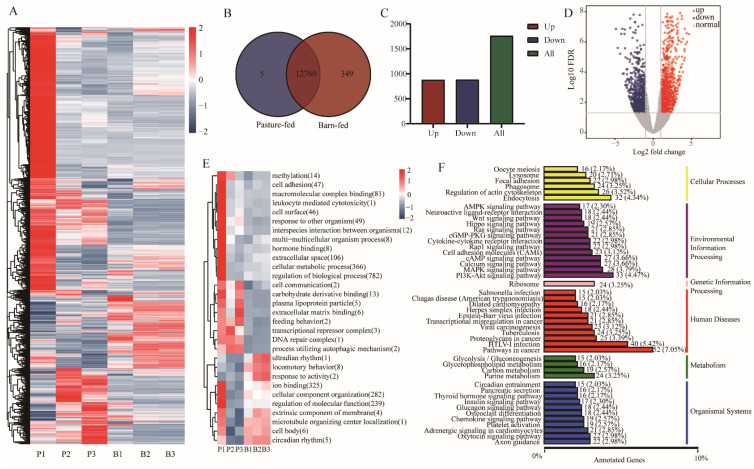
mRNA analysis of muscles from pasture-fed and barn-fed black goats. (**A**) Hierarchical cluster analysis of genes identified in the pasture-fed and barn-fed group. (**B**) Number of genes expressed in the two groups. (**C**) Number of differentially expressed genes. (**D**) Distribution of the genes expressed in the two groups, red dots and blue dots indicate genes significantly up-regulated or down-regulated in the barn-fed group. (**E**) GO Enrichment analysis of the differentially expressed genes. (**F**) KEGG analysis of the differentially expressed genes.

**Figure 3 foods-11-00381-f003:**
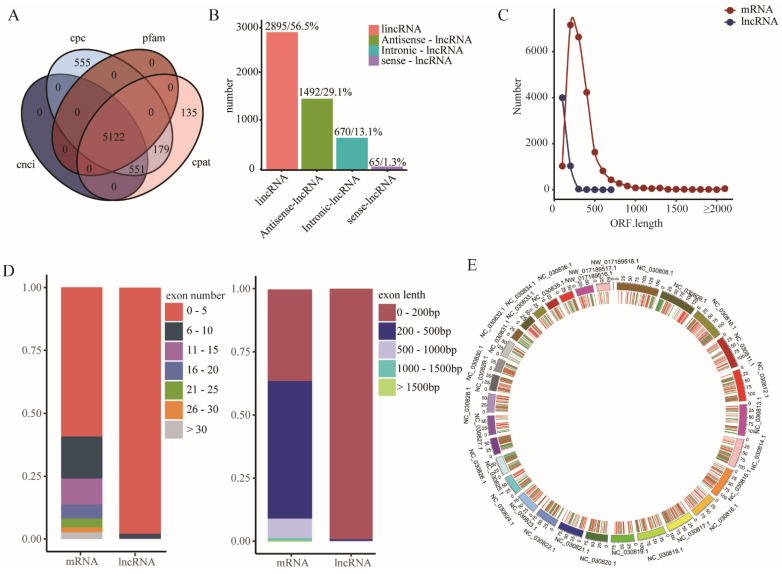
Identification of lncRNAs in goat muscles. (**A**) Prediction method Venn diagram. (**B**) Sta-tistics of the number of four different types of lncRNAs. (**C**) ORF length distribution of mRNA and lncRNA. (**D**) Statistical of exon number and length. (**E**) Distribution of the identified lncRNA and mRNA in the chromosome.

**Figure 4 foods-11-00381-f004:**
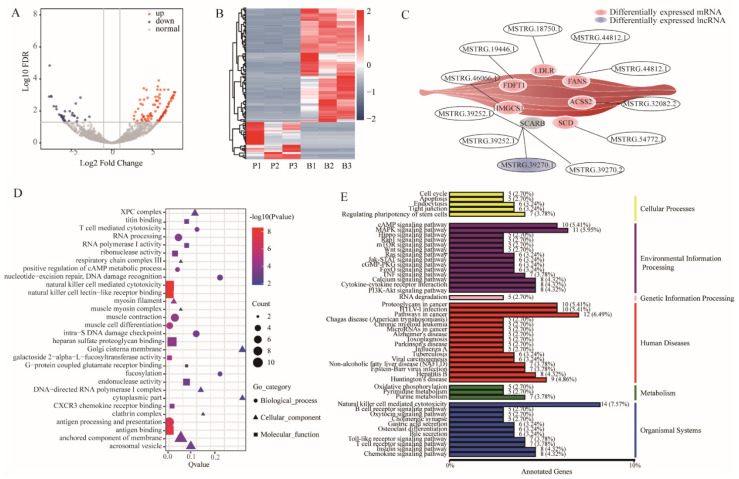
Differential expression and enrichment analysis of lncRNAs. (**A**) Differentially expressed lncRNAs in the muscles. (**B**) Cluster diagram of differentially expressed lncRNA. (**C**) Relationship between lncRNAs and the target genes. (**D**) GO analysis of biological process, molecular function and cell components of differentially expressed lncRNA CIS target genes among samples. (**E**) Classification of differentially expressed lncRNA CIS target gene KEGG.

**Figure 5 foods-11-00381-f005:**
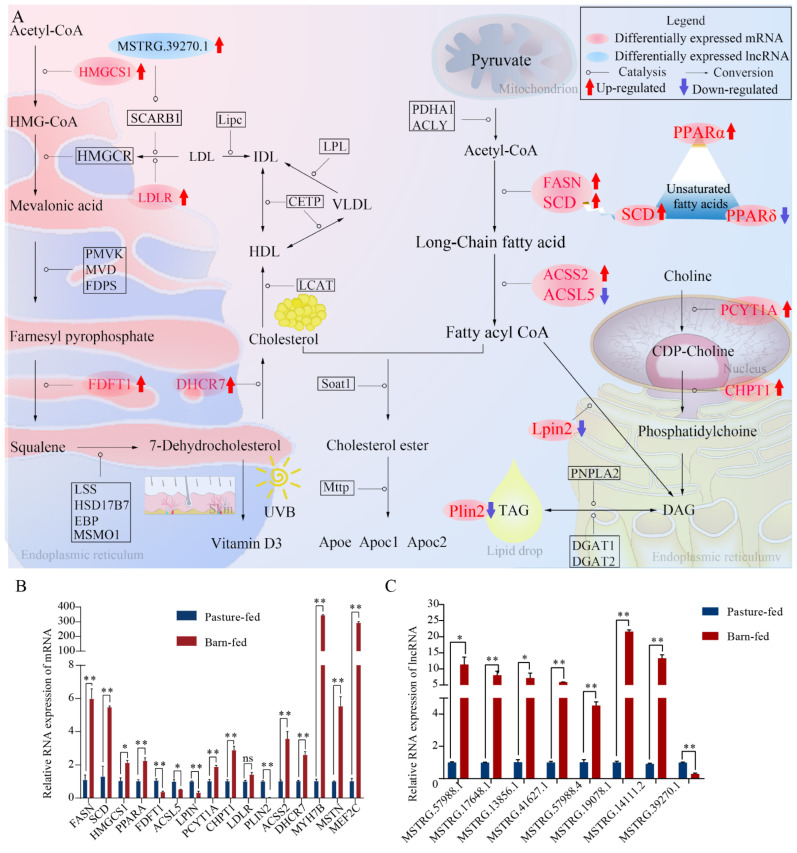
Analysis of the underlying molecular mechanism and qRT-PCR analysis of the differentially expressed genes and lncRNAs. (**A**) Analysis of the underlying molecular mechanism of the two feeding models. Genes and lncRNAs in red or blue indicate the differentially expressed genes or lncRNAs identified in this study; arrows in red or blue indicate up-regulated or down-regulated in the barn-fed group. (**B**) qRT-PCR analysis of the differentially expressed mRNA. (**C**) qRT-PCR analysis of the differentially expressed lncRNA. * and ** indicate significant different (*p* < 0.05 or *p* < 0.01).

**Table 1 foods-11-00381-t001:** Fatty acid analysis of the longissimus lumborum from pasture-fed and barn-fed goats.

	Content in Different Feeding Model (%)			Content in Different Feeding Model (%)	
Fatty Acid	Pasture-Fed (*n* = 3)	Barn-Fed (*n* = 3)	Fatty Acid%	Pasture-Fed (*n* = 3)	Barn-Fed (*n* = 3)
(C 6:0)	0.600 ± 0.077 ^A^	0.130 ± 0.054 ^B^	(C 16:1) cis	2.960 ± 0.173 ^A^	1.230 ± 0.119 ^B^
(C 8:0)	0.084 ± 0.009	0.100 ± 0.030	(C 17:1) cis	1.240 ± 0.079	2.900 ± 0.649
(C 10:0)	0.199 ± 0.019	0.176 ± 0.024	(C 18:1n9c) trans	32.200 ± 1.329	0.620 ± 0.615
(C 12:0)	0.090 ± 0.011 ^b^	0.160 ± 0.017 ^a^	(C 18:1n9t) cis	2.450 ± 0.797	52.680 ± 0.553
(C 13:0)	0.000 ± 0.000 ^b^	0.070 ± 0.011 ^a^	(C 20:1) cis	0.040 ± 0.042	0.040 ± 0.007
(C 14:0)	2.050± 0.179 ^B^	3.360 ± 0.182 ^A^	(C 24:1) cis	0.000 ± 0.000 ^b^	0.130 ± 0.046 ^a^
(C 15:0)	0.890 ± 0.082 ^b^	1.620 ± 0.145 ^a^	(C 18:2n6t) trans	0.300 ± 0.010	0.488 ± 0.223
(C 16:0)	25.060 ± 0.928 ^b^	54.730 ± 9.748 ^a^	(C 18:2n6c) cis	1.210 ± 0.311	0.360 ± 0.054
(C 17:0)	2.770 ± 0.128	3.840 ± 0.541	(C 18:3n6) cis	0.040 ± 0.039	0.250 ± 0.041
(C 18:0)	27.370 ± 1.357	22.350 ± 11.160	(C 18:3n3) cis	0.000± 0.000	0.180 ± 0.076
(C 20:0)	0.050 ± 0.050	0.026 ± 0.008	(C 20:3n6) cis	0.000± 0.000	0.090 ± 0.075
(C 21:0)	0.080 ± 0.076	0.160 ± 0.027	(C 20:4n6) cis	0.000± 0.000	0.050 ± 0.024
(C 23:0)	0.000 ± 0.000 ^B^	0.070 ± 0.006 ^A^	(C 20:5n3) cis	0.000± 0.000 ^b^	0.030 ± 0.008 ^a^
(C 14:1) cis	0.320 ± 0.027 ^b^	0.450 ± 0.024 ^a^	(C 20:2) cis	0.000± 0.000	0.050 ± 0.027
(C 15:1) cis	0.000 ± 0.000	0.070 ± 0.055	Total FA	100.000 ± 0.003	100.100 ± 15.790
PUFA n-3	0.000 ± 0.000	0.100 ± 0.048	PUFA n-6	0.660 ± 0.228 ^a^	0.270 ± 0.062 ^b^
MUFA	39.210 ± 1.843	42.810 ± 14.970	PUFA	1.550 ± 0.314	1.280 ± 0.245
Total SFA	59.250 ± 2.065 ^b^	68.790 ± 2.603 ^a^	Total UFA	40.760 ± 0.021 ^a^	31.770 ± 0.086 ^b^
AI	23.530 ± 4.907	41.990 ± 9.237	NVI	2.490 ± 0.143 ^A^	1.360 ± 0.183 ^B^
OFA	27.110 ± 1.106 ^b^	39.920 ± 2.655 ^a^	TI	2.700 ± 0.226	3.940 ± 2.691
DFA	65.680 ± 1.257 ^a^	18.070 ± 11.550 ^b^			

Note: ^a, b^ for *p* < 0.05 and ^A, B^ for *p* < 0.01. Abbreviations: cis indicates the chemical structural formula of the fatty acid is cis-form.; SFA, saturated fatty acid; UFA, unsaturated fatty acid; MUFA, monounsaturated fatty acid; PUFA, poly-unsaturated fatty acid; NVI, nutritive value index; AI, atherogenic index; TI, thrombogenic index; OFA describes dietary fatty acids that have adverse hypercholesterolemic effects on humans; DFA dietary fatty acids having a desirable neutral hypocholesterolemia effect in humans.

## Data Availability

Sequences and metadata generated in this work are deposited at the Sequence Read Archive (https://www.ncbi.nlm.nih.gov/sra) under accession number PRJNA784758 [48].

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
