# Peer review of "Novel Insights into the Differences in Nutrition Value, Gene Regulation and Network Organization between Muscles from Pasture-Fed and Barn-Fed Goats"

_foods, 2022, doi:10.3390/foods11030381_

Round 1

Reviewer 1 Report

The aim of the research was to determine the influence of housing system and nutrition for nutritional value, physicochemical features, meat microstructure, serum biological indexes and genetic determinants of meat quality of black goats. The number of goats (n = 3) used in the experiment is relatively small, test methods used are correct. The chapter "Introduction" provides an overview of the current world literature on the topic of the article. The discussion is well carried out and exhausting. References well chosen but the way they are listed needs to be revised.  The paper requires additions and corrections. The list of proposed changes is given below:

General comments:

Please prepare the article according to the instructions for the authors.

  • L7-12 Add initialize the first and last names which to be used in the "Author Contributions" section
  • In the Author Contributions section, use initials and surname instead of full name
  • Use "p" in italic for significance
  • The table 1 and S1-S5 headings must be in bold (all words)
  • Before all numbers in parentheses, add spaces
  • Abbreviated name journal must be regular non-block letters
  • The volume number must be in italic
  • In the References section, for a range of pages, use the long "-" from the insert function for all References items

Detailed comments:

L14 Give the aim of the research as the first sentence,

L17 selenium or magnesium?

L17 add description of significant differences for the basic chemical composition, physicochemical features, serum biochemical indexes of the goats.

L82 + add information about the type of barn (with windows?) type of floor, density, environmental parameters (temperature, humidity, light day length) for pasture and barn system

L96 + add information about the method of measuring electrical conductivity, describe in more detail the method of determining shear force, fatty acid content, measuring acidity of meat (pH of calibration buffers)

L179 add table or figure

Table 1 introduce the notations a, b for p < 0.05 and A, B for p < 0.01

Why is the sum of SFA and UFA greater than 100% (Pasture-fed group = 100.1% and for Barn-fed 100.49%), what was the method of determining FA (content FA)?

Captive or Barn-fed? in the headline

Figure 2

Figure 2C and 2 D under the figure 2 A and 2 B. The current form is not legible - explanations are too small letters

Table S2 chemical instead of biochemical

pH instead of PH, Electrical conductivity instead of conductivity?

Table S3 in the title Amino acid content (in mg( in 100 g of meat (longissimus lumborum) of black goats

Table S4 Mineral contents (in mg) in 1 kg of meat (longissimus lumborum) of black goats

Author Response

Dear editors and reviewers:

Thank you for your letter and for the reviewers’ comments concerning our manuscript. It is our great honor for our manuscript to be reviewed and considered to be revised. These comments are all valuable and very helpful for improving our manuscript, as well as the important guiding significance to our further researches. We have studied all comments carefully and revised our manuscript word by word. All of the suggestions have been revised and highlighted in the manuscript. We hope that the revised manuscript will qualify enough to be accepted by the reviewers and editors. The responses to the comments are listed in the following point-by-point.

Response to Reviewer 1:

  1. L7-12 Add initialize the first and last names which to be used in the "Author Contributions" section.

Reply: Thanks for your reminder. We have added the initials of the first and last names in the section.

  1. In the Author Contributions section, use initials and surname instead of full name.

Reply: We have replaced the full name into initials in the Author Contributions section.

  1. Use "p" in italic for significance.

Reply: We have revised this problem in the paper. (Please see in line 160, 205 and 275).

  1. The table 1 and S1-S5 headings must be in bold (all words)

Reply: We have revised the heading of table 1 and S1-S5 into bold. (please see in line 204 and supplementary materials Table S1-S5).

  1. Before all numbers in parentheses, add spaces

Reply: We have added spaces in all of the numbers in parentheses in the paper.

  1. Abbreviated name journal must be regular non-block letters. The volume number must be in italic. In the References section, for a range of pages, use the long "-" from the insert function for all References items.

Reply: We have revised this problem in the paper.

  1. L14 Give the aim of the research as the first sentence.

Reply: We have added the aim of this research as the first sentence in the paper as below. The physiological and biochemical characters of muscles derived from pasture-fed or barn-fed black goats were detected and RNA-seq was performed to reveal the underlying molecular mechanism that how the pasture feeding affected the nutrition and flavor of the meat.

  1. L17 selenium or magnesium? L17 add description of significant differences for the basic chemical composition, physicochemical features, serum biochemical indexes of the goats.

Reply: We have added the description of significant differences in the paper as below. We found that the branched chain amino acids, unsaturated fatty acids and zinc in the muscle of pasture-fed goats were significantly higher than those in the barn-fed group, while the heavy metal elements, cholesterol and low-density lipoprotein cholesterol were significantly lower.

  1. L82 + add information about the type of barn (with windows?) type of floor, density, environmental parameters (temperature, humidity, light day length) for pasture and barn system

Reply: We have added the information that you listed in the paper as below. The barn-fed goats were fed in attic houses with double pitched roof. The leaky floor was more than 1.8 m above the ground. There is a rear window in every 100 m2 of the house, and the total daylighting area of the window is more than 1 / 20 of the total area. Each sheep covers an area of 0.8 ~ 1.2 m2. As this research was performed in winter, the temperature in the goat house is about 10-25 ℃, the humidity is about 60 % ~ 70 %, and the illumination time is about 10 hours. For the pasture-fed goats, they were fed in the natural environment that can move and seek food and water freely. We randomly selected 3 goats from the barn-fed and pasture-fed group respectively for the following study. The figures below show the environments of pasture-fed and barn-fed goats.

  Figure (A) pasture-fed environment; (B) barn-fed environment

  1. L96 + add information about the method of measuring electrical conductivity, describe in more detail the method of determining shear force, fatty acid content, measuring acidity of meat (pH of calibration buffers)

Reply: We have added the information that you listed in the paper as below. The muscle samples were obtained immediately after the slaughtering and the meat color data were read with the meat color meter (MATTHAUS, Germany). The muscle conductivity was detected using the carcass meat conductivity tester LF-STAR (MATTHAUS, Germany). The shear force was measured with C-LM3 digital display muscle tenderness instrument (MATTHAUS, Germany) according to the instrument's instructions. In brief, the muscle samples with length × width × height more than 6 cm × 3 cm × 3 cm were prepared and the fascia and fat were removed. The samples were then put into 80 ℃ water bath until the center temperature reached 70 ℃. The shear force was measured after cooling with cold water. The determination of fatty acids was carried out in accordance with the national food safety standard determination of fatty acids in foods (GB 5009.168-2016) issued by the State Food and Drug Administration (http://down.foodmate.net/wap/index.php?moduleid=23&itemid=50488). In brief, the muscle sample is firstly hydrolyzed by hydrochloric acid, and the fat extract is collected. The fat is then saponified and the fatty acid is methylated, and the fatty acid is finally determined by gas chromatography.

The acidity of the meats was measured using a pH meter with potassium hydrogen phthalate solution (pH = 4.0) and mixed phosphate buffer (pH = 6.8) as the calibration buffers. (please see in line 104-135).

  1. L179 add table or figure

Reply: We have added the statistical chart of muscle fiber area as table S2, and adjusted the order of the tables accordingly.

  1. Table 1 introduce the notations a, b for p < 0.05 and A, B for p <0.01.

Reply: We have replaced all of the notations indicating the significant differences in the paper according to your comments.

  1. Why is the sum of SFA and UFA greater than 100% (Pasture-fed group = 100.1% and for Barn-fed 100.49%), what was the method of determining FA (content FA)?

Reply: Each of the FA was determined according to the methods of national food safety standard determination of fatty acids in foods (GB 5009.168-2016). While the content of total FA, total SFA, total UFA, MUFA and PUFA was obtained by arithmetic sum of the relevant fatty acids. As the value of each fatty acid content is rounded to 3 decimal places, there is error in calculating the summation.

  1. Captive or Barn-fed? in the headline

Reply: This is a mistake spell and it should be Barn-fed here. We have revised this in the paper.

  1. Figure 2C and 2 D under the figure 2 A and 2 B. The current form is not legible - explanations are too small letters

Reply: Thanks for your kindly suggestion. We have revised this figure in the paper to make it more legible. (please see in figure 2).

  1. Table S2 chemical instead of biochemical. pH instead of PH, Electrical conductivity instead of conductivity? Table S3 in the title Amino acid content (in mg ( in 100 g of meat (longissimus lumborum) of black goats. Table S4 Mineral contents (in mg) in 1 kg of meat (longissimus lumborum) of black goats

Reply: Thanks for your kindly reminders. We are so sorry for these errors and have revised all of them in the paper.

Reviewer 2 Report

This study is an interesting investigation of the nutrition value of black goat meat. However, some parts of the manuscript need clarification, as follows:

  • at the very introduction to the results (L165) - the terms "pasture-fed" and "barn-fed" are introduced to denote samples whose results are further processed and clarified. But then in Figure 1A and Table 1 the term "Captive" appears. Is there any special reason for this change, if so - clarify it.
  • In figure 1 appear some symbols which needs to be explained in the caption of the figure (as **; *; ns)
  • in tables 1, S2-S5 are used signs as "↑ and ↓ indicate significantly higher or lower", but lower or higher from which value?
  • Also in tables presented in the Supplement material, you explain that "* and ** indicate significant different (p < 0.05 or p < 0.01)", but in the mentioned tables are used also "***" and "****" which are not explained
  • L151: when you mention the  2(-Delta Delta C(T)) method, to ensure the reproducibility of your work - be sure to provide a reference that provides further clarification of this method or you should clarify it in more detail
  • some of the references are incomplete (without the indicated pages or number of the papers) - see references as no. 19; 22; 23; 25; 42; 44
  • the conclusion that something is "better" than something else (L429) must be argued. It must be added - what are the quality, flavour and nutrition indicators that support this statement. 
  • Sincerely

Author Response

Dear editors and reviewers:

Thank you for your letter and for the reviewers’ comments concerning our manuscript. It is our great honor for our manuscript to be reviewed and considered to be revised. These comments are all valuable and very helpful for improving our manuscript, as well as the important guiding significance to our further researches. We have studied all comments carefully and revised our manuscript word by word. All of the suggestions have been revised and highlighted in the manuscript. We hope that the revised manuscript will qualify enough to be accepted by the reviewers and editors. The responses to the comments are listed in the following point-by-point.

Response to Reviewer 2:

1.This study is an interesting investigation of the nutrition value of black goat meat. However, some parts of the manuscript need clarification, as follows: at the very introduction to the results (L165) - the terms "pasture-fed" and "barn-fed" are introduced to denote samples whose results are further processed and clarified. But then in Figure 1A and Table 1 the term "Captive" appears. Is there any special reason for this change, if so - clarify it.

Reply: Thanks for your kindly reminders. This is a mistake spell and it should be Barn-fed here. We have revised this in the paper.

  1. In figure 1 appear some symbols which needs to be explained in the caption of the figure (as **; *; ns) in tables 1, S2-S5 are used signs as "↑ and ↓ indicate significantly higher or lower", but lower or higher from which value? Also, in tables presented in the Supplement material, you explain that "* and ** indicate significant different (p < 0.05 or p < 0.01)", but in the mentioned tables are used also "***" and "****" which are not explained.

Reply: Thanks for your kindly reminders. It is our oversight that we didn’t proofread the notations of the significant difference in the tables. We have replaced all of the notations indicating the significant differences in the paper as below. a, b for p < 0.05 and A, B for p <0.01.

  1. L151: when you mention the 2 (-Delta Delta C(T)) method, to ensure the reproducibility of your work - be sure to provide a reference that provides further clarification of this method or you should clarify it in more detail

Reply: Thanks for your suggestion. We have supplied a reference (Title: Analysis of Relative Gene Expression Data Using Real-Time Quantitative PCR and the 2−ΔΔCT Method) that provides further clarification of this method in the paper.

  1. some of the references are incomplete (without the indicated pages or number of the papers) - see references as no. 19; 22; 23; 25; 42; 44

Reply: Thanks for your kindly reminders. We have supplied the missing information in the references.

  1. Through the systematic determination of minerals, amino acids, fatty acids and serum biochemical indexes in goat muscle, our study showed that the quality, flavor and nutritional value of muscles derived from pasture-fed is better than barn-fed black goats. the conclusion that something is "better" than something else (L429) must be argued. It must be added - what are the quality, flavour and nutrition indicators that support this statement.

Reply: Thanks for your suggestion. We delated the description about something is "better" than something else. Instead, we just point out that something in one group is different with the other group.
